# Multifamily Determination of Phytohormones and Acidic Herbicides in Fruits and Vegetables by Liquid Chromatography–Tandem Mass Spectrometry under Accredited Conditions

**DOI:** 10.3390/foods9070906

**Published:** 2020-07-09

**Authors:** Ángel Grande Martínez, Francisco Javier Arrebola Liébanas, Rosario Santiago Valverde, María Elena Hernández Torres, Juan Ramírez Casinello, Antonia Garrido Frenich

**Affiliations:** 1Department of Chemistry and Physics (Analytical Chemistry Area), Agrifood Campus of International Excellence ceiA3, University of Almería, E-04120 Almería, Spain; angelquimico@yahoo.es (Á.G.M.); arrebola@ual.es (F.J.A.L.); 2Bio-Clinical Analytical Laboratory (LAB), PITA (Almería Science and Technology Park, Albert Einstein 7, E-04131 Almería, Spain; rsantiago@lab-sl.com (R.S.V.); ehernan@lab-sl.com (M.E.H.T.); jramirez@lab-sl.com (J.R.C.)

**Keywords:** phytohormones, acidic herbicides, fruits and vegetables, multifamily method, UHPLC–MS/MS

## Abstract

A 7-min multifamily residue method for the simultaneous quantification and confirmation of 8 phytohormones and 27 acidic herbicides in fruit and vegetables using ultra high-performance liquid chromatography (UHPLC) coupled to tandem mass spectrometry (MS/MS) was developed, validated according to SANTE 12682/2019, and accredited according to UNE-EN-ISO/IEC 17025:2017. Due to the special characteristics of these kinds of compounds, a previous step of alkaline hydrolysis was carried out for breaking conjugates that were potentially formed due to the interactions of the analytes with other components present in the matrix. Sample treatment was based on QuEChERS extraction and optimum detection conditions were individually optimized for each analyte. Cucumber (for high water content commodities) and orange (for high acid and high water content samples) were selected as representative matrices. Matrix-matched calibration was used, and all the validation criteria established in the SANTE guidelines were satisfied. Uncertainty estimation for each target compound was included in the validation process. The proposed method was applied to the analysis of more than 450 samples of cucumber, orange, tomato, watermelon, and zucchini during one year. Several compounds, such as 2,4-dichlorophenoxyacetic acid (2,4-D), 4-(3-indolyl)butyric acid (IBA), dichlorprop (2,4-DP), 2-methyl-4-chlorophenoxy acetic acid (MCPA), and triclopyr were found, but always at concentrations lower than the maximum residue level (MRL) regulated by the EU.

## 1. Introduction

Phytohormones play essential roles in the regulation of physiological processes in plants and vegetables, most of them involved in the plants’ growth, development, defense, and response to environmental stimuli [1,2]. In consequence, plant hormones have an influence on plant development and crop yield, directly or indirectly. Therefore, research into the hormone physiology of plants has become an important target for agriculture development [3,4]. Plant hormones can be considered one of the cornerstones of molecular breeding and a key to opening the door of modern agriculture. They may be generated by plants in a natural way. However, in the last few years, the artificial synthesis of plant hormones and compounds with similar functions (plant growth regulators, PGRs) in vegetables has increased [5] because these compounds are capable of increasing harvest grain productivity and improving agricultural production. However, special care and control must be taken over them since it has been shown that, in some cases, they have carcinogenic or teratogenic effects on animals and humans. For these reasons, the control and analysis of these compounds have a special interest in food safety [6]. Like other chemicals used in farming, their usage is restricted and must be carried out responsibly, assuring that the limits established by the mandatory regulations are not exceeded [7,8].

On the other hand, pesticides are used worldwide to control pests and diseases in agriculture, allowing an improvement in product quality and crop productivity. The presence of pesticide residues in food has become common and necessary to maintain the supply needs of the current population. In consequence, the control of these substances acquires relevant importance, even more so when adverse health effects that this type of residue can have in the long term have been demonstrated. In the case of acidic herbicides, their acid properties make them possess special characteristics of reactivity, increasing their possibility of interacting with other natural components present in plants and being transformed into conjugated residues by biological reactions. Conjugate formation is conditioned by a large number of factors, such as the specific stage during crop development in which these compounds are applied, if applied as free acids, salts, or esters, and the climatic conditions. For these reasons, the European Union (EU) has included them in residue definitions and, therefore, it regulates them for maximum residue levels (MRLs), not only for the active ingredients but also for other possible structures that can be formed due to chemical interactions with natural compounds or secondary reactions. Hence, it is important to control these analytes and their related compounds because, on some occasions, they can be more harmful than the parent compound [9]. The control of their residues in fruits and vegetables minimizes the health risks associated with their consumption or the environmental damage that may result.

It is easy to find monofamily methods for the determination of phytohormones (auxins [10,11], cytokinins [12,13], gibberellins [14,15]) or methods specifically developed for the analysis of acidic herbicides [16,17,18] in the literature. Some scientific publications have even jointly analyzed various families of phytohormones [19,20,21,22,23,24]. It is rare to find studies in which phytohormones and acidic herbicides are grouped together under one method [25,26] The issue of the residue determination of analytes that belong to different functional families is the broad range of physical–chemical properties and structures that these compounds present. Often, specific characteristics of the target analytes cause the extraction procedure to be modified to make it adequate for those particular properties. In this particular case of study, a previous alkaline treatment is necessary due to the trend of conjugate formation as a consequence of the interaction of the target compounds with matrix components. The goal of the alkaline hydrolysis is to convert the conjugates into the parent compound by breaking residues through hydrolysis of the sample [27]. Many of the existing methods hardly manage to gather each and every one of the principles on which “standard” QuEChERS methodology is based [24,28]. Most of the methods found require a high volume of solvents [29] or complex sample treatments such as derivatization [30] or solid-phase extraction (SPE) [31]. The present work achieves the processing of a large number of samples in a short time (quick), to be carried out in a simple way (easy), without using large quantities of solvents and reagents (cheap), completely and quantitatively extracting the amount of target analytes present in the samples (effective), being able to withstand procedural variations (rugged), and being reliable (safe).

The high separation efficiency of chromatographic systems and its ability to be combined with different detectors make it the most useful technique for the analysis of phytohormones and acidic herbicides [30,31]. The analysis of these compounds is carried out almost exclusively by liquid chromatography (LC), although some references proposing the use of gas chromatography (GC) [32] can also be found. LC coupled to mass spectrometry (MS) provides a powerful tool to analyze both phytohormones [4,22] and acidic herbicides [33] in food matrices. Furthermore, LC can be coupled to other detection systems due to the presence of chromophores of the analytes under UV–vis conditions, such as diode arrays (DADs) ultraviolet (UV) [34,35] and fluorescence detectors (FLDs), [36], but these have less applicability in this field due to their technical limitations.

The developed analysis method allows the simultaneous determination of phytohormones and acidic herbicides in the same “run” and in only 7 min. Cucumber and orange have been taken as representative commodities of fruits and vegetables containing high water content and high acid and high water content, respectively, because they are the groups of matrices more commonly cultivated in southeastern Spain. In addition, with the aim of increasing the applicability of the method, matrices such as tomato, watermelon, and zucchini were also checked, obtaining results that meet those validation criteria established by the European SANTE guidelines [37]. The method was developed in order to reach quantification limits that were sufficiently low enough to allow the determination of concentrations of such compounds at trace levels in order to evaluate the compliance with current food safety regulations. The validation of the method has been carried out, including the uncertainty of the method [37,38], an important parameter when a result is utilized to decide whether it indicates compliance or noncompliance with a specification of regulatory limits. In most publications, it is a parameter that is not estimated.

The aim of the present study is the development, optimization, and validation of a method for the simultaneous determination of 27 acidic herbicides and 8 phytohormones in cucumber and orange matrices at trace levels, applying a modified QuEChERS extraction method and ultra-high performance liquid chromatography coupled to tandem mass spectrometry (UHPLC–MS/MS) determination. The proposed method has demonstrated proper reliability and robustness in order to fulfill accreditation criteria under UNE-EN-ISO/IEC 17025:2017 and its application to routine conditions in a laboratory for food safety monitoring.

## 2. Materials and Methods

### 2.1. Reagents and Chemicals

A commercial phytohormone standard of gibberellic acid (GA) was purchased from Riedel de Haën (Seelze-Hannover, Germany). Indole-3-acetic acid (IAA), 2,4-dichlorophenoxyacetic acid (2,4-D), naphthylacetic acid (NAA) and naphthylacetamide (NA) were supplied by Fluka (Sleeze-Hannover, Germany), while 4-chlorophenoxy acetic acid (4-CPA), 4-chloro-2-methyl-phenoxy acetic acid (MCPA), 4-chloro-2-methyl-phenoxy butyric acid (MCPB), and N^6^-benzyladenine (BA) were obtained from Sigma-Aldrich (Steinheim, Germany).

Acidic herbicide reference standards were purchased from Dr. Enhrenstofer (Augsburg, Germany), Riedel-de-Haën (Sigma-Aldrich), Fluka, Chem Service (West Chester, PA, USA), and HPC Standards GmbH (Borsdorf, Germany). Triphenyl-phosphate (TPP), used as an internal standard, was purchased from Supelco (Bellefonte, PA, USA).

Individual analyte stock standard solutions of each compound were prepared in the range of 200 to 300 mg/L of concentrations, considering standard purity, by accurately weighing powder or liquid of individual analytical standards into 50 mL volumetric flasks and dissolving them with acetone (LC–MS gradient grade solvent from Sigma-Aldrich), except for fenoprop [2,4,5-TP] in acetonitrile (HPLC grade, Sigma Aldrich). For that, an analytical balance AB204-S from Mettler Toledo (Greifensee, Switzerland) and a vortex mixer Heidolph (Kelheim, Germany) Model Reax 2000 were used. These solutions were stored at −20 °C in the dark. All the stock solutions were not stored for more than 6 months in order to avoid stability problems. From these solutions, various working standard solutions at a concentration of 10 mg/L of each compound were prepared weekly by appropriate dilution with acetonitrile and stored in screw-capped glass tubes at −20 °C in the dark.

Methanol (LC–MS gradient grade), sodium citrate dihydrate, sodium citrate dibasic sesquihydrate, and sodium hydroxide were supplied by Sigma-Aldrich. Anhydrous sodium chloride (99.5%) and magnesium sulfate (97%) were purchased from Panreac (Barcelona, Spain). Methanol and highly purified water (Millipore, Bedford, MA, USA) were used for sample preparation and mobile phases. Formic acid (>99.0% Optima, LC–MS grade), acetic acid (purity higher than 99.8% for HPLC) and sulfuric acid (96% solution in water, extra pure) were obtained from Fluka.

### 2.2. Sample Preparation

Samples were obtained from supermarkets in Almería (southeastern Spain). Sampling was made in accordance with Directive 2002/63/EC. A representative portion of the sample was homogenized using a Sammic SK-3 kitchen blender and a Reax 2 rotary agitator from Heildoph (Schwabach, Alemania), processed, and the analysis was completed on the day of sample reception. If samples were not analyzed immediately, they were stored for the shortest possible time in a freezer (−20 °C) until analysis. Organic samples of cucumber and orange were purchased from specialized stores in Almería (Spain). These samples, showing the absence of the target analytes, were used as blanks for the preparation of calibration standards and for recovery and precision studies during the method validation.

### 2.3. Extraction Procedure and Sample Analysis

Briefly, 10 ± 0.1 g of homogenized fruit or vegetable sample was weighed into a 50-mL polypropylene tube, and 5 mL of Milli-Q water was added in high water content samples (or 10 mL in high acid content and high water content samples). The mixture was shaken and homogenized for 2 min. Then, 10 mL of acetonitrile acidified with 1% formic acid and containing TPP at 0.05 mg/L as a surrogated internal standard was added. Afterwards, alkaline hydrolysis was performed, adding 300 µL of sodium hydroxide 5 N for 30 min and heating at 80 °C in a thermostated water bath. The mixture was shaken in a vortex for 2 min and neutralized with 300 µL of sulphuric acid 5 N. Then, 4 g of anhydrous MgSO_4_, 1 g of NaCl, 1 g of dehydrated sodium citrate, and 0.5 g of sodium citrate dibasic sesquihydrate were added. The mixture was shaken vigorously for 2 min and centrifuged (centrifuge from Orto Alresa, Mod. Cónsul, Madrid, Spain) at 3060 × *g* for 10 min. Finally, 750 µL of the supernatant was transferred to a chromatographic vial and diluted with 750 µL of Milli-Q water. If necessary, the extract was filtered through a 0.20-µm nylon filter for the removal of potential solid interferents before injection into the UHPLC system.

On the other hand, internal quality control was used in each analysis sequence in order to ensure the method’s suitability and the truthfulness of the results obtained. For this purpose, the following samples were analyzed in each batch: (i) a reagent blank to check that there were no reagent interferences, and a blank matrix to test if the matrix was free of target compounds, (ii) a matrix-matched calibration, (iii) a spiked blank sample at the LOQ (10 µg/Kg) to evaluate if recoveries were between 70% and 120%, and (iv) a control sample at 5x LOQ concentration (50 µg/Kg), injected every 25 samples, in order to check the precision of the method and to verify that the instrument was stable throughout the sequence. The maximum admissible error for the control sample was ±20% of spiked concentration.

### 2.4. UHPLC–MS/MS

Instrumental determination was done using an Agilent 1290 Infinity UPLC system (Santa Clara, CA, USA) coupled with an AB Sciex Triple QuadTM 5500 mass spectrometer (Foster City, CA, USA). The chromatographic separation was performed using an Acquity UPLCTM BEH C-18 column (100 × 2.1 mm id, 1.7 µm particle size) from Waters (Mildford, MS, USA). Analyst software version 1.6 (AB Sciex, USA) was used for data acquisition and processing.

A total of 10 µL of the sample was injected at a flow rate of 0.35 mL/min. The temperature of the chromatographic column was set at 30 °C. Chromatographic analyses were carried out using 1% acetic acid and 5% methanol in water (eluent A) and 1% acetic acid in methanol (eluent B) as mobile phases. The gradient elution started with 10% eluent B, which was linearly increased up to 90% in 4.0 min. This composition was held for further 0.5 min before returning to the initial conditions in 0.5 min, followed by a re-equilibration time of 2 min, to give a total run time of 7 min.

The mass analyses were performed with an ESI source using scheduled multiple reaction monitoring (MRM), with rapid switching between positive (ESI^+^) and negative (ESI^−^) modes and with N_2_ as the nebulizer. The switching time was 50 ms. The MRM detection window was 40 s, with a 5.0 ms pause. The target scan time was 0.25 s, and the mass spectrometric resolution was 1 Da.

The parameters of the MS source employed were electrospray ionization voltage (IS) ± 4500 V (depending on the ESI+ or ESI− mode), source temperature 500 °C, air curtain gas pressure (CUR) 40 psi, ion source gas 1 (GS1) and 2 (GS2) 55 psi, and collision gas pressure (CAD) 7 psi. Curtain gas was nitrogen (>95% purity), and the gas used to fragment the precursor ions (collision gas) was argon (99.9999%).

### 2.5. Validation

The developed method was validated according to the European SANTE guidelines (SANTE 12682/2019). The calculated validation parameters were retention time window (RTW), specificity, linearity, trueness, precision, limit of quantification (LOQ), and uncertainty.

RTW: Defined as the average retention time ± six standard deviations of the retention time (RT ± 6SD), with a tolerance of ±0.1 min [37]). RTW values were calculated by analyzing 10 blank samples spiked at 50 µg/kg.Specificity: Responses for reagent blanks and blank control samples had to be less than 30% LOQ.Linearity and working range: Linearity was studied in the range of 10 to 150 µg/Kg using matrix-matched standard calibration to overcome the matrix effect. The determination coefficient (r2) must be higher than 0.98 for all the studied compounds, and deviation of the residuals of each calibration point must be in the range of ±20%.Trueness: Expressed as the mean recovery in %, it was evaluated by spiking blank samples (*n* = 10) at two different spiking levels (10 and 50 µg/Kg); values must be in the range 70–120%. Recovery of TPP was also checked, and values must be in the range 70–130%.Precision study: Repeatability (intraday precision) and intermediate precision (interday precision) data were calculated at the same concentrations tested for trueness (10 and 50 µg/Kg). Intraday precision data were obtained from the analysis of spiked blank samples (*n* = 10) on the same day and by the same analyst, while interday precision values were obtained over ten different days by three different analysts. In both cases, the obtained values must be lower than or equal to 20%, expressed as relative standard deviation (RSD).Limit of Quantification: LOQ was established as the lowest spike level meeting recoveries in the range 70–120% and precision values lower or equal to 20%.Expanded uncertainty: Expanded uncertainty (U) was estimated based on intralaboratory validation data for individual analytes contained in the target matrices (cucumber and orange) at two concentration levels (10 and 50 μg/Kg, respectively); *n* = 10. In order to simplify the uncertainty estimation (u’), u´_Precision_ and u´_bias_ were considered as main contributor variabilities. They included the uncertainty associated with the precision method and the uncertainty associated with the preparation of standards and the trueness of the method, respectively. Calculations were based on Equation (1).

(1)$$u´=\sqrt{u{´}_{Precision}{}^{2}+u{´}_{bias}{}^{2}}$$

A coverage factor of 2 (k = 2, level of confidence = 99.54%) was applied to calculate the expanded uncertainty (U = ku´). U-values must be ≤50% for the LOQ concentration and ≤40% for concentrations higher than or equal to 50 µg/kg.

## 3. Results and Discussion

### 3.1. Optimization of UHPLC–MS/MS

The first step considered in the method development was the optimization of the mass spectrometer parameters. Standard solutions of each individual analyte were infused using negative and positive ionization modes (ESI^−^ and ESI^+^, respectively). The possibility of using the two ionization modes during the execution of the method resulted in an advantage due to the differences between the families of the analyzed compounds (phenoxy acid, benzoic acid, imidazolinones), obtaining better sensitivity using the optimum ionization mode for each analyte. During this optimization, the goal was to find a compromise between the highest abundance precursor/product ion combinations and the *m*/*z* ratio in order to obtain high sensitivity but, at the same time, high selectivity. The rest of the experimental parameters of the ionization source were set up to obtain adequate ionization of the compounds and the proper volatilization of the mobile phase. The mass analyzer parameters were adjusted in order to achieve the optimal ion transitions that allow the detection of the compounds at low concentrations but, at the same time, avoid potential interferences of the studied matrices. The mass spectrometric parameters for each compound are listed in Table 1.

For the chromatographic separation of the compounds, methanol was present in both proposed phases. Due to the characteristics of the target analytes, the addition of methanol was necessary to achieve an adequate chromatographic profile. In addition, acetic acid was added in both phases because low pH (apparent values ≤5.5) values prevent the dissociation of the acid compounds and the ionization of residual silanol groups in the stationary phase, avoiding peak tailing and slightly improving the resolution between chromatographic peaks. The total chromatographic resolution between the compounds was not reached (Figure 1), although the observed coelutions were solved through the spectral resolution provided by the mass analyzer.

In these conditions, the total run time to determine the target analytes was 7 min. This analysis time is lower than the published methods for a lower or similar number of these compounds [27,30]. Specifically, in [27], 14 analytes were determined in 8 min, while in [30], the separation of 28 analytes needed 22 min.

### 3.2. Optimization of Extraction Method

In the present study, the QuEChERS extraction method was adapted to extract the target analytes. The main modifications with regard to other methodologies previously described were (i) acidification of the extraction solvent for avoiding potential dissociation of the analytes and improving the method precision, (ii) customized addition of water to samples (5 mL of water for cucumber or 10 mL of water for orange) in order to adapt water content to the original acidity of the samples, (iii) addition of a lower amount of both sodium hydroxide 5 N (pH ≈ 12) and sulfuric acid 5 N (pH ≈ 1) to reduce potential contamination of the samples, reduce costs, and miniaturize the method, and (iv) carrying out hydrolysis for 30 min but increasing temperature from 40 to 80 °C in order to improve the robustness of the method and the reliability of the results. In addition, a reduced matrix effect was observed due to the 2-fold dilution of the obtained extracts. It represented an improvement with regard to alternative clean-up steps with sorbents (better selectivity) and avoided analyte losses by increasing extraction efficiency for most of the target analytes (enhanced robustness).

With the optimal extraction conditions described in the experimental section, the proposed method was tested, analyzing 10 replicates of spiked blank samples at 50 µg/Kg of target compounds. Table 2 and Table 3 show the results obtained for cucumber and orange, respectively. All the studied compounds were recovered with rates between 70% to 120%, and precision data were always ≤20%. The results complied with the limit values set by the European SANTE.

### 3.3. Method Validation

The validation protocol was designed in order to fulfill the requirements and obtain accreditation according to UNE-EN-ISO/IEC 17025:2017 by ENAC (National Accreditation Body in Spain) and SANTE guidelines for reliable identification and quantification of phytohormones and acidic herbicides. Methods found in the literature were generally validated, but it is very unusual to find a method accredited by an international quality standard.

The identification of target compounds was based on RTW values. Table 2 and Table 3 show the obtained values, meeting the threshold (±0.1 min) established by SANTE guidelines for all compounds. Analytes were confirmed by mass spectrometry by comparing the ion intensity ratios of their most characteristic ions with those obtained for standards analyzed at similar concentrations. In all cases, the obtained values were in the permitted tolerance range (±30%) for confirmation as a compound.

#### 3.3.1. Specificity

Specificity was investigated by analyzing ten blank samples and checking that no interfering chromatographic peaks were observed at the same RTWs of the analytes. Figure 2 and Figure 3 show the absence of interferences in the blanks of cucumber and orange, respectively. Hence, no matrix interferences or other analytes that would cause a false-positive signal were observed at the RTW of each analyte.

#### 3.3.2. Linearity and Working Range

The matrix-matched calibration curves were prepared at 10, 25, 50, 100, and 150 µg/kg. This working range was established for all compounds, taking into account that there was good linearity, and it included the MRLs in the target matrices. Weighting least-squares regression was used, plotting peak area versus concentration of the calibration standards, with weighting factor 1/x, and not forcing curves to pass through the origin. A weighted fit of the calibration line was used to compensate for the observed homoscedasticity and to improve the accuracy of the analytical results.

The experimentally obtained correlation coefficient (r^2^) was always higher than 0.98 for all target analytes. The individual residual of each point of the calibration curve did not deviate more than ±20% from their values predicted, complying with the requirements established by EU legislation. With the fit of both parameters, the capacity of the calibration function could be assured as adequate within the concentration range studied.

#### 3.3.3. Trueness (Trueness Assessment)

To ensure that the developed method provided truthful results, the % recovery for each analyte at the concentration of 10 µg/Kg was also calculated. Recoveries ranged from 86% to 115% in cucumber samples (Table 2), and from 73% to 120% in orange commodities (Table 3), according to the requirements of the UNE-EN-ISO/IEC 17025:2017 accreditation and SANTE guidelines.

#### 3.3.4. Precision study

Intraday precision values, ranging between 1–20% for cucumber (Table 2) and between 1–17% for orange (Table 3) samples, were obtained. Interday precision values were in the range 5–20% for cucumber (Table 2) and 4–20% for orange (Table 3) samples. It can be observed that RSDs for intra- and interday precision studies were always equal or lower than 20%.

#### 3.3.5. Limit of Quantification

LOQs were set at 10 µg/kg for all analytes. Additionally, at this concentration level, the signal-to-noise ratio (S/N) of the chromatographic peaks was calculated, verifying that the target analytes originated an S/N higher than 10 for the smallest transition ion (qualifier). It is remarkable that all the analytes had a LOQ lower or equal to the lowest MRL established in each matrix.

#### 3.3.6. Uncertainty

The U-values obtained for the lowest level of concentration studied (10 µg/kg) in cucumber samples ranged from 18% to 46%, with an average of 30%, while for the higher studied level (50 μg/kg), it ranged from 10% to 38% (Table 2), with an average of 23%. For orange samples (Table 3), the U-values were between 15% to 44% (28% average) and 7% to 37% (17% average) for the lowest and highest concentration levels, respectively. These values were in agreement with EU requirements, where a generalized U-budget of ±50% is applicable as the default value [37,39].

### 3.4. Sample Analysis

To improve the scope of the method, new matrices such as tomato, melon, and zucchini were also verified. Studies (*n* = 3) of trueness and precision at 10 µg/Kg were carried out. Tomato and watermelon were checked with calibration curves prepared with cucumber, whereas for zucchini, orange was used as the representative matrix.

A total of 457 samples were analyzed with this method during the last year. Real samples of cucumber (67), orange (150), tomato (120), watermelon (60), and zucchini (60) were analyzed. A total of 32 positive samples were found in tomato and orange samples; no positives were detected in cucumber, melon, or zucchini samples (Table 4). Tomato (6 positive samples—1.3% of the total samples) was the only food commodity with target analytes above their MRL in the group of matrices with high water content, whereas, for the high acid and high water content samples, positive results were only found in the representative matrix orange (26 positive samples—5.7% of the total sample analyzed). In all cases, the presence of target compounds was detected as below the MRLs established by the EU.

2,4-dichlorophenoxyacetic acid (2,4-D), which can act as herbicide or phytohormone, and 4-(3-indolyl)butyric acid (IBA), a phytohormone, were detected in the positive samples of tomato. In orange, four analytes were detected. Two of those compounds can be used as herbicides or phytohormones: 2,4-D and dichlorprop (2,4-DP) (Figure 4). The other two detected compounds can be used as herbicides: 2-methyl-4-chlorophenoxy acetic acid (MCPA) and triclopyr. In tomato, the analyte most found was IBA, detected in four of the studied samples (66.7% of the total positive samples) at a concentration that ranged from 12 to 52 µg/Kg. The most commonly detected analyte in orange was MCPA, being determined in 10 of the studied samples (38.5% of the total positive samples) at a concentration that ranged from 13 to 74 µg/Kg.

The compounds with values higher than the LOQ (positives samples) were identified and confirmed as described in Section 3.3, fulfilling the requirements based on RTW values and the ratio between quantification ion and confirmation ion intensity.

## 4. Conclusions

The developed method allows us to group in a single “run” the phytohormones and acidic pesticides widely used in cucumber (a matrix with high water content) and orange (a matrix with high acid and water content) by UHPLC–MS/MS. The extraction procedure was based on the QuEChERS method but with some modifications to adapt the extraction to the broad physicochemical properties of the target analytes. The instrumental analysis time was 7 min. The proposed method has been designed to be successfully implemented in testing laboratories to perform routine analyses, thanks to its simple sample treatment and rapid chromatographic analysis.

To ensure method suitability, a validation (linearity, specificity, trueness, precision, LOQs, and uncertainty) was performed in compliance with the SANTE 2019 guidelines. Uncertainty is not a very common parameter found in the bibliography about organic contaminants in fruits and vegetables. It has been included in the validation process in order to carry out an evaluation of compliance with the requirements of current legislation about MRLs established by the EU.

Subsequently, 457 real samples were analyzed. Compounds such as 2.4-D, IBA, 2.4-DP, MCPA, and triclopyr were the only five compounds found, often at concentrations lower than their MRLs. It should be noted that most of the previous publications did not carry out an extensive application of the method to real samples, and the detection of positive cases have rarely been reported. The results obtained show evidence of its applicability to the analysis of real samples in routine residue monitoring programs and that it is fit-for-purpose.

## Figures and Tables

**Figure 1 foods-09-00906-f001:**
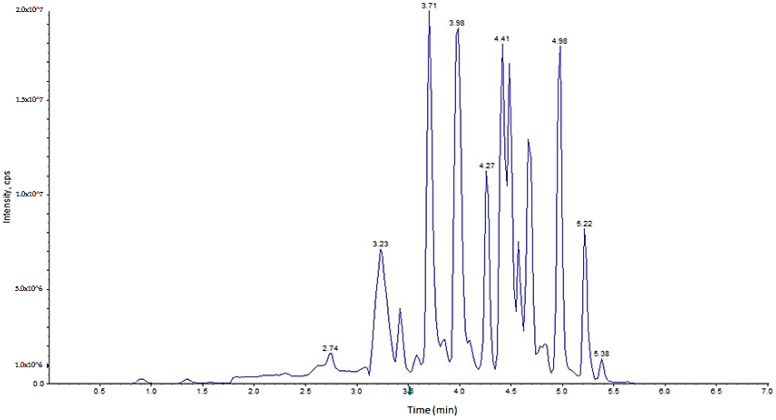
Total ion chromatogram (TIC) of a spiked cucumber sample at 150 µg/kg.

**Figure 2 foods-09-00906-f002:**
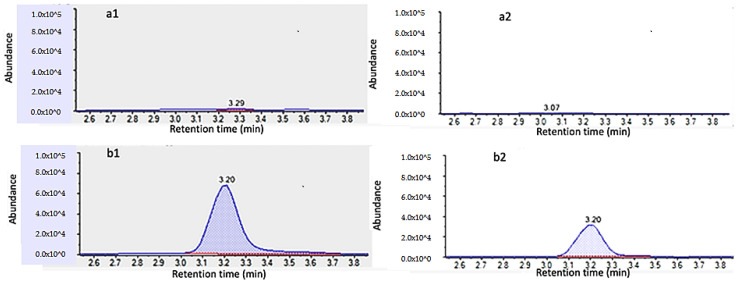
UHPLC–MS/MS chromatogram of (**a**) blank cucumber sample, monitoring transitions at *m/z* 222/141 (**a1**) and *m/z* 222/114 (**a2**), and (b) blank cucumber sample spiked with 10 µg/kg of quinmerac, monitoring transitions at m/z 222/141 (**b1**) and m/z 222/114 (**b2**).

**Figure 3 foods-09-00906-f003:**
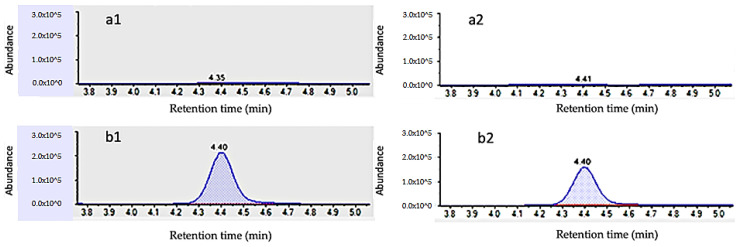
UHPLC–MS/MS chromatogram of (**a**) blank orange sample, monitoring transitions at m/z 219/161 (**a1**) and m/z 220/163 (**a2**), and (**b**) blank orange sample spiked with 10 µg/kg of 2,4-D, monitoring transitions at m/z 220/161 (**b1**) and m/z 220/163 (**b2**).

**Figure 4 foods-09-00906-f004:**
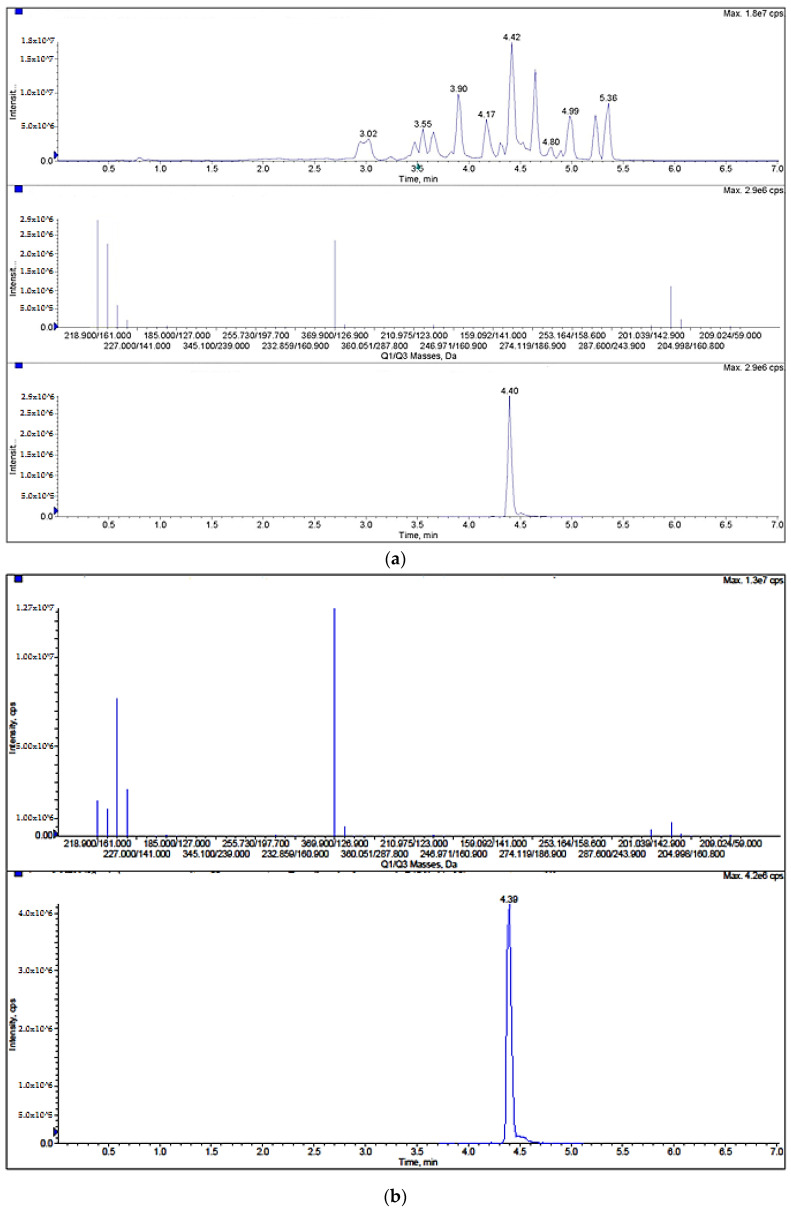
(**a**) Positive orange sample containing 2,4 D (total ion chromatogram, relation ions, and UHPLC–MS/MS chromatogram), and (**b**) relation ions and UHPLC–MS/MS chromatogram of a matrix-matched standard of 2,D.

**Table 1 foods-09-00906-t001:** Shows the specific detection conditions for each analyte.

	Precursor Ion (m/z)	Product Ion (m/z)		Declustering Potential (DP) (V)	Entrance Potential (EP) (V)	Collision Energy (CE) (V)	Collision Cell Exit Potential (CXP) (V)
Naphthylacetamide	186.1	141.2	^a^	3.5	1	10	29
	186.1	115.2	^b^	3.5	1	10	10
2,4,5-trichlorophenoxyacetic acid [2,4,5-T]	253.2	158.6	^a^	4.6	−55	−10	−46
	253.2	194.4	^b^	4.6	−55	−10	−13
2,4-Dichlorophenoxyacetic acid [2,4-D]	218.9	161.0	^a^	4.3	−65	−10	−20
	220.9	163.0	^b^	4.3	−65	−10	−14
2,4-dichlorophenoxy butyric acid [2,4-DB]	247.0	160.9	^a^	4.7	−35	−10	−14
	247.0	124.8	^b^	4.7	−35	−10	−9
2-(4-Chlorophenoxy)acetic acid [4-CPA]	185.0	127.0	^a^	3.8	−55	−10	−20
	187.0	128.8	^b^	3.8	−100	−10	−7
4-(3-indolyl)butyric acid [IBA]	203.0	186.0	^a^	3.8	41	10	19
	203.0	130.0	^b^	3.8	41	10	12
Gibberellic acid	345.1	239.0	^a^	2.8	−60	−10	−20
	345.1	227.2	^b^	2.8	−150	−10	−10
Bentazon	239.0	132.0	^a^	3.8	−100	−10	−38
	239.0	175.0	^b^	3.8	−100	−10	−4
2-naphthyloxyacetic acid (BNOA)	201.0	142.9	^a^	4.1	−100	−10	−40
	201.0	115.0	^b^	4.1	−100	−10	−17
Bromoxynil	275.7	80.9	^a^	4.1	−40	−10	−30
	275.7	78.9	^b^	4.1	−40	−10	−9
Clomazone	240.0	125.0	^a^	4.3	66	12	30
	240.0	89.0	^b^	4.3	66	10	6,5
Dicamba	219.0	174.8	^a^	3.6	−5	−10	−8
	221.0	177.0	^b^	3.6	−5	−10	−9
Dichlorprop [2,4-DP]	232.9	160.9	^a^	4.5	−25	−10	−18
	232.9	124.9	^b^	4.5	−25	−10	−13
Fenoprop [2,4,5-TP]	269.0	196.8	^a^	4.8	−70	−10	−14
	269.0	160.9	^b^	4.8	−70	−10	−15
Fenoxaprop P	362.0	288.0	^a^	5.2	126	10	25
	362.0	119.0	^b^	5.2	126	10	18
Flamprop	320.1	121.0	^a^	4.4	−75	−10	−22
	320.1	247.7	^b^	4.4	−75	−10	−45
Fluazifop	328.0	254.0	^a^	4.5	126	10	35
	328.0	282.0	^b^	4.5	126	10	16
Fluroxypyr	253.1	194.8	^a^	3.6	−120	−10	−18
	253.1	232.8	^b^	3.6	−120	−10	−13
Haloxyfop	360.1	287.8	^a^	4.9	−95	−10	−20
	360.1	195.8	^b^	4.9	−95	−10	−13
Haloxyfop-etoxyl	434.1	315.9	^a^	5.2	11	12	25
	434.1	288.0	^b^	5.2	121	10	6,5
Haloxyfop-methyl	376.0	316.0	^a^	5.1	131	10	30
	376.0	288.0	^b^	5.1	126	10	12
Imazamox	306.1	261.1	^a^	3.0	71	10	25
	306.1	245.9	^b^	3.0	71	10	28
Imazapyr	274.1	186.9	^a^	3.0	−30	−10	−18
	274.1	230.0	^b^	3.0	−30	−10	−13
Imazethapyr	287.6	243.9	^a^	3.4	−75	−10	−18
	287.6	186.1	^b^	3.4	−75	−10	−21
Ioxynil	369.9	126.9	^a^	4.3	−90	−5	−25
	369.9	116.0	^b^	4.3	−90	−5	−10
2-methyl−4-chlorophenoxy acetic acid (MCPA)	199.0	141.0	^a^	4.3	−65	−10	−20
	201.0	143.0	^b^	4.3	−65	−10	−12
2-methyl−4-chlorophenoxy butyric acid (MCPB)	227.0	141.0	^a^	4.7	−55	−10	−20
	229.0	143.0	^b^	4.7	−55	−10	−12
Mecoprop (MCPP)	212.9	140.9	^a^	4.6	−45	−10	−18
	212.9	70.9	^b^	4.6	−45	−10	−9
Quimerac	222.1	141.1	^a^	2.9	36	10	45
	222.1	114.1	^b^	2.9	36	10	8
Quinclorac	241.9	223.9	^a^	3.5	26	10	21
	241.9	161.0	^b^	3.5	26	10	16
Sulcotrione	328.9	139.1	^a^	3.5	111	10	25
	328.9	111.1	^b^	3.5	130	10	10
Triclopyr	255.7	197.7	^a^	4.5	−15	−10	−14
	255.7	217.8	^b^	4.5	−15	−10	−11

^a^ Quantification ion. ^b^ Confirmation ion.

**Table 2 foods-09-00906-t002:** Retention time windows and validation parameters for the target compounds in cucumber.

Compound	Repeatability				Intermediate Precision				Uncertainty		RTW
	Rec (%) ^a^	RSD (%) ^a^	Rec (%) ^b^	RSD (%) ^b^	Rec (%) ^a^	RSD (%) ^a^	Rec (%) ^b^	RSD (%) ^b^	(%) ^a^	(%) ^b^	(min.)
Gibberellic acid	109	6	75	4	93	17	85	6	28	14	2.75–2.85
Imazamox	101	3	116	3	97	19	103	15	46	27	2.95–3.15
Imazapyr	115	6	71	2	100	14	82	8	25	18	2.98–3.08
Quimerac	109	3	94	4	108	13	81	18	25	32	3.14–3.24
Imazethapyr	114	9	117	6	107	16	103	15	29	27	3.38–3.48
Quinclorac	104	4	99	4	103	19	86	12	37	21	3.46–3.56
Naphthylacetamide	92	5	120	2	90	18	111	7	35	14	3.47–3.57
Sulcotrione	95	4	102	2	94	12	90	10	24	18	3.7–3.57
Fluroxypyr	114	6	91	3	94	17	86	9	28	17	3.52–3.62
Dicamba	97	6	86	5	82	17	82	14	30	27	3.56–3.66
2-(4-Chlorophenoxy)acetic acid (4-CPA)	102	4	82	4	88	13	82	11	22	22	3.73–3.83
Bentazon	102	6	108	1	89	10	93	9	18	16	3.73–3.83
4-(3-indolyl)butyric acid (IBA)	86	14	120	4	93	20	92	20	42	33	3.78–3.88
2-naphthyloxyacetic acid (BNOA)	104	6	89	3	91	13	85	7	23	12	4.02–4.12
Bromoxynil	108	6	109	2	94	13	92	10	23	16	4.02–4.12
Clomazone	96	6	118	4	84	17	116	8	30	10	4.22–4.32
2-methyl-4-chlorophenoxy acetic acid (MCPA)	107	8	92	3	85	18	86	15	28	28	4.24–4.34
Ioxynil	110	10	83	4	87	19	86	8	30	17	4.26–4.36
Flamprop	101	14	103	11	92	18	96	14	32	26	4.28–4.38
2,4-Dichlorophenoxyacetic acid (2,4-D)	99	7	99	3	92	13	88	11	24	19	4.34–4.44
Triclopyr	105	9	101	6	92	13	89	13	23	24	4.37–4.47
Fluazifop	107	7	94	2	104	20	99	14	38	29	4.38–4.48
Dichlorprop (2,4-DP)	103	9	103	4	94	15	95	12	28	22	4. 45–4.55
Mecoprop (MCPP)	105	7	101	4	87	15	93	11	25	20	4.48–4.58
2,4,5-trichlorophenoxyacetic acid (2,4,5-T)	113	10	102	4	103	20	89	12	36	22	4.54–4.64
2,4-dichlorophenoxy butyric acid (2,4-DB)	110	11	102	16	113	18	85	20	38	33	4.63–4.73
2-methyl-4-chlorophenoxy butyric acid (MCPB)	106	12	107	11	105	16	90	14	33	23	4.66–4.76
Fenoprop (2,4,5-TP)	113	7	105	4	93	14	91	11	24	20	4.75–4.85
Haloxyfop	102	11	79	7	98	20	89	17	39	38	4.79–4.89
Haloxyfop-methyl	107	14	101	11	92	14	84	20	24	33	4.95–5.05
Haloxyfop-etoxyl	104	19	118	13	85	19	76	19	32	24	5.07–5.17
Fenoxaprop P	97	20	119	10	87	20	92	19	35	34	5.10–5.20

Rec: Recovery. RSD: Relative standard deviation. ^a^ Level concentration in the validation study of 10 µg/kg. ^b^ Level concentration in the validation study of 50 µg/kg.

**Table 3 foods-09-00906-t003:** Retention time windows and validation parameters for the target compounds in orange.

Compound	Repeatability				Intermediate Precision				Uncertainty		RTW
	Rec (%) ^a^	RSD (%) ^a^	Rec (%) ^b^	RSD (%) ^b^	Rec (%) ^a^	RSD (%) ^a^	Rec (%) ^b^	RSD (%) ^b^	(%) ^a^	(%) ^b^	(min.)
Imazamox	90	1	109	2	86	8	98	9	15	16	2.96–3.06
Quimerac	73	2	81	2	80	14	77	14	30	27	3.12–3.24
Naphthylacetamide	108	2	120	2	101	10	107	6	20	10	3.47–3.57
Quinclorac	85	1	83	1	91	10	78	12	21	23	3.47–3.57
Sulcotrione	95	4	106	1	93	12	91	9	24	16	3.47–3.57
Fluroxypyr	101	6	112	6	99	10	98	8	19	14	3.53–3.63
Bentazon	96	6	120	11	100	10	97	4	21	7	3.74–3.84
2-(4-Chlorophenoxy)acetic acid (4-CPA)	88	3	104	6	89	15	85	13	30	21	3.75–3.85
4-(3-indolyl)butyric acid (IBA)	101	2	120	3	106	13	105	9	27	16	3.79–3.89
2-naphthyloxyacetic acid (BNOA)	80	4	112	8	91	19	94	6	44	10	4.03–4.13
Bromoxynil	103	4	117	7	106	13	104	5	27	9	4.03–4.13
Clomazone	119	1	119	2	114	8	109	6	15	11	4.23–4.33
2-methyl-4-chlorophenoxy acetic acid (MCPA)	83	7	109	4	91	19	94	9	42	16	4.25–4.35
Ioxynil	117	4	114	6	118	11	108	8	22	16	4.28–4.38
Flamprop	107	9	106	3	113	12	111	6	25	13	4.29–4.39
2,4-Dichlorophenoxyacetic acid (2,4-D)	78	8	107	8	91	17	93	5	39	9	4.36–4.46
Triclopyr	94	9	98	4	102	13	103	8	29	17	4.38–4.48
Dichlorprop (2,4-DP)	84	8	115	5	91	16	99	8	35	13	4.46–4.56
Mecoprop (MCPP)	91	8	114	4	88	20	98	8	40	15	4.49–4.59
2,4,5-trichlorophenoxyacetic acid (2,4,5-T)	92	5	115	7	99	12	98	8	26	14	4.55–4.65
2,4-dichlorophenoxy butyric acid (2,4-DB)	106	2	111	9	102	18	107	19	35	37	4.64–4.74
2-methyl-4-chlorophenoxy butyric acid (MCPB)	90	6	120	7	100	14	106	12	31	21	4.66–4.76
Fenoprop (2,4,5-TP)	110	5	117	5	107	10	94	10	19	15	4.76–4.86
Haloxyfop	119	17	107	8	119	11	109	9	20	18	4.90–5.00
Haloxyfop-methyl	120	3	113	1	107	11	92	11	19	17	4.95–5.05
Haloxyfop-etoxyl	107	14	106	12	112	14	86	13	30	22	5.08–5.18
Fenoxaprop P	105	14	118	11	110	19	93	18	40	28	5.10–5.20

Rec: Recovery. RSD: Relative standard deviation. ^a^ Level concentration in the validation study of 10 µg/kg. ^b^ Level concentration in the validation study of 50 µg/kg.

**Table 4 foods-09-00906-t004:** Compounds in analyzed samples above the LOQ of the method expressed as mg/kg.

Compound Detected	Tomato		Orange	
	MRL (mg/kg)	Concentration (mg/kg)	MRL (mg/kg)	Concentration (mg/kg)
2,4-Dichlorophenoxyacetic acid (2,4-D)	0.01	0.012 to 0.016	1	0.014 to 0.670
Dichlorprop (2.4-DP)	0.05	Not detected	0.3	0.014 to 0.097
2-methyl-4-chlorophenoxy acetic acid (MCPA)	0.05	Not detected	0.05	0.013 to 0.074
Triclopyr	0.01	Not detected	0.1	0.017 to 0.066
4-(3-indolyl)butyric acid (IBA)	0.1	0.012 to 0.052	0.1	Not detected

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
