# Peer review of "Multifamily Determination of Phytohormones and Acidic Herbicides in Fruits and Vegetables by Liquid Chromatography–Tandem Mass Spectrometry under Accredited Conditions"

_foods, 2020, doi:10.3390/foods9070906_

Round 1

Reviewer 1 Report

The work presented by Angel Grande Martínez and colleagues is interesting and quite ambitious in attempting to determine many analytes of interest, showing an important application in food safety. 

The manuscript can be improved when a linguistic help would be obtained to improve the understanding of the work. 

The method section has not been clearly presented; thus it is difficult how the experiments were carried out while developing and optimising the method.  

In the ‘Results and discussion’ section authors have described the protocols instead of presenting and discussion the data. Protocols and experiments are part of the methods 

Introduction 

There are some linguistic issues related to grammar and expression that needs to be corrected and addressed. Some examples are shown below: 

L 37 what does a development controller refer to? 

L 49 legal mandatory regulation can be simply expressed as mandatory regulation(s) or is it a legislation? Please clarify and be specific 

L 91 Please correct and rewrite the sentence, as diode array detectors are based on chromophores od the analytes under the UV/Vis condition. 

Methods 

L 124-130 How long were the stock standard solutions stored in a freezer? Were the concentrations of standards verified before use, due to the fact that some analytes became unstable in methanol. 

A multi-compound working standard solutions could be better expressed as various working standard solutions. Were the working standards prepared fresh before use? Storage of working standards would introduce a systematic error in quantification of analytes. 

L 133-134 please clarify the sentences and delete any repetitions 

L 143-147 Instrument used during the study can be incorporated in the protocols rather than listed out here. 

L 153-157 The text needs to be re-written. Which organic samples were purchased and then used as blank matrices? How similar were the blank matrices with the actual samples? 

Spiked samples for the recovery study would be a better term used instead of fortified samples, 

L 158-173 Extraction protocols have not been clearly written and could be misleading. Which samples were extracted, please specify. How many replicates were prepared for extraction? The speed of the centrifuge must be expressed in g force instead of rpm 

L 195-219 This section must be re-written to improve understanding, because validation is important before quantitative analysis. If this was not done properly, the quantitation could not be accurately performed. 

Results and discussion 

Figure 1. The TIC shows a poor resolution, which is also mentioned in the text. Please explain whether attempts were made to extend the gradient from 10% to 90% in order to obtain a better baseline resolution 

Mass spectra must be presented, showing standards and samples. 

Were attempts made in comparing candidates of mobile phase system? 

L 249-270 The section was also not clear, and rewriting is needed for a better understanding. 

L 285 Not fully captured what was meant here. Do authors mention that the product ion from each analyte have a product ion and a confirmation ion for identification and quantitation? Please show the confirmation ions as they are not displayed in Table 1. 

Explain and justify the use of internal standard in helping the quantification.  

How do the chromatogram and mass spectra of samples look like? Any matrix interferences? 

L 293 use consistent units for concentrations throughout the manuscript. Explain the role of internal standard here. Linearity commonly used the LOQ as the lowest concentration. Please explain. 

Trueness in assessing the recovery. Please clarify whether blank matrices or real  samples were used for the study. 

L 310-319 The section describes protocol and should be placed in methods 

What are the LOQ values for each analyte? What are the LOD values? 

L 363 Measured samples had levels below the regulatory limits. Explain and justify the accuracy of the measurements, as samples were only detected and were not able to be quantified.  

Author Response

Introduction 

There are some linguistic issues related to grammar and expression that needs to be corrected and addressed. Some examples are shown below: 

CommentL 37 what does a development controller refer to? 

Response: The use of plant hormones is useful for controlling the physiological development of plants along the different plant growth stages (sprout, seedling, vegetative, budding, flowering, ripening). According to reviewer’s indication, the sentence has been rephrased in order to simplify and clarify it (L38-39, new version).

Comment: L 49 legal mandatory regulation can be simply expressed as mandatory regulation(s) or is it a legislation? Please clarify and be specific 

Response: According to reviewer’s indication, the term “legal” has been deleted in the body of the text because it can result redundant.

CommentL 91 Please correct and rewrite the sentence, as diode array detectors are based on chromophores od the analytes under the UV/Vis condition. 

Response: The sentence has been rephrased according to the reviewer’s suggestion (L92-93 new version).

 Methods 

Comment: L 124-130 How long were the stock standard solutions stored in a freezer? Were the concentrations of standards verified before use, due to the fact that some analytes became unstable in methanol. 

Response: The stability of the solutions was evaluated and warranted for at least six months in the described storage conditions. The sentence “All the standard solutions were not stored more than 6 months in order to avoid stability problems” has been added to clarify it (L138, new version).

 Comment: A multi-compound working standard solutions could be better expressed as various working standard solutions. Were the working standards prepared fresh before use? Storage of working standards would introduce a systematic error in quantification of analytes.

Response: According to the reviewer’s suggestion, various working standard solutions have been used (L139, new version). These working standard solutions were prepared weekly and stored in the freezer at -20ºC. This has been indicated in the new version of the manuscript (L140-141, new version).

 Comment: L 133-134 please clarify the sentences and delete any repetitions 

Response: This sentence has been deleted in order to clarify and avoid repetitions. The information on the quality of highly purified water has been indicated in the next sentence in the manuscript: “Methanol and highly purified water (Millipore, Bedford, MA, USA) were used for the sample preparation and mobile phases”, (L145-147, new version).

Comment: L 143-147 Instrument used during the study can be incorporated in the protocols rather than listed out here. 

Response: The apparatus and instrument section has been deleted in the new version of the manuscript. This information has been incorporated in the protocols, according to the referee´s suggestion.

Comment: L 153-157 The text needs to be re-written. Which organic samples were purchased and then used as blank matrices? How similar were the blank matrices with the actual samples? 

Response: The text has been rewritten according to the referee´s suggestion: Organic samples of cucumber and orange were purchased from specialized stores in Almería (Spain). These samples showing the absence of the target analytes were used as blanks for the preparation of calibration standards and for recovery and precision studies in the method validation (L167-169, new version).

In terms of organoleptic and physical properties, organic and actual samples collected during the study did not present significant differences. According to the content of target analytes, organic samples did not present any target compound above the limit of detection of the proposed method, whereas real samples in some cases contained some of the studied compounds (as described in the manuscript in the section of analysis of real samples).

Comment: Spiked samples for the recovery study would be a better term used instead of fortified samples, 

Response: The referee is right and the correct term is “spiked” instead of “fortified”. This terms has been revised throughout the manuscript.

Comment: L 158-173 Extraction protocols have not been clearly written and could be misleading. Which samples were extracted, please specify. How many replicates were prepared for extraction? The speed of the centrifuge must be expressed in g force instead of rpm 

Response: All the extraction protocol has been reviewed and clarified, according to the referee´s suggestion (L174-186, new version). The speed of the centrifuge has been expressed in g force (L183, new version).

On the other hand, number of replicates used has been indicated in the 2.5 (L236, 250, 255,265) and 3.3 (L368, new version)sections of the manuscript.

Comment: L 195-219 This section must be re-written to improve understanding, because validation is important before quantitative analysis. If this was not done properly, the quantitation could not be accurately performed. 

Response: This section has been re-phrased in accordance to the referee´s recommendation (L345-445, new version).

Results and discussion 

CommentFigure 1. The TIC shows a poor resolution, which is also mentioned in the text. Please explain whether attempts were made to extend the gradient from 10% to 90% in order to obtain a better baseline resolution.

Response: Different chromatographic conditions, including the suggested by the referee and with other pH modifiers as formic acid and ammonium formate or using acetonitrile instead of methanol, were evaluated in order to improve resolution between chromatographic peaks with poor results. Not significant differences were observed regard those described in the proposed method. As indicated in the text, the use of MS/MS working in MRM mode provided the required spectral resolution to determine target compounds successfully. This has also allowed the development of a quick method (run times of 7 minutes) which is very important in routine analysis laboratories.

Comment: Mass spectra must be presented, showing standards and samples. 

Response:  In the revised version of the manuscript a new Figure (Figure 4) has been added, showing chromatograms and mass spectra of 2,4-D in a standard and an orange sample.

Comment: Were attempts made in comparing candidates of mobile phase system? 

Response: As indicated above, different mobile phases were tested and the results regarding sensitivity and peak shape were similar. This mobile phase was chosen because it is the one recommended in acid herbicide methods by European Residue Laboratories and because of the nature of the compounds, which are generally more stable in an acidic environment

Comment: L 249-270 The section was also not clear, and rewriting is needed for a better understanding. 

Response: The section has been rewritten in order to clarify it (L307-329, new version).

Comment: L 285 Not fully captured what was meant here. Do authors mention that the product ion from each analyte have a product ion and a confirmation ion for identification and quantitation? Please show the confirmation ions as they are not displayed in Table 1. 

Response: The sentence has been rewritten to clarify it (L356-359, new version). Yes, each analyte is identified and confirmed by tandem mass spectrometry monitoring the two most intense and characteristic product ions obtained from the precursor ion under the mass spectrometric conditions described in Table 1. Both ions, quantification and confirmation, were included in Table 1 and differentiated at the bottom of the table with the super index a for the quantification ion and b for the confirmation ion.

On the other hand, the information in line 285 simply shows that since all the compounds under study have been characterized by a precursor ion and two product ions, the criteria of identification points indicated in the European Commission Decision 2002/657 / EC are met. However, since it may be confusing, it has been removed in the revised version of the manuscript.

Comment: Explain and justify the use of internal standard in helping the quantification.  

Response: The internal standard has been added before the extraction process to the sample test portion, including that used for the calibration standards, to account for sources of error throughout the extraction step in the method. Recovery rates of TPP  between 70 and 130% indicated adequate extraction of the target compounds. In consequence, if an acceptable recovery was obtained for TPP, this would indicate that the extraction process has performed well, and consequently also the correct extraction of the target compounds.

Comment: How do the chromatogram and mass spectra of samples look like? Any matrix interferences? 

Response: Due to the high selectivity of the mass spectrometric conditions selected, no interferences were observed in the sample blanks as observed in the new Figures 2 and 3.

Comment: L 293 use consistent units for concentrations throughout the manuscript. Explain the role of internal standard here. Linearity commonly used the LOQ as the lowest concentration. Please explain. 

Response: Units have been reviewed throughout the manuscript to harmonize them. Linearity started at the LOQ as the lowest concentration, as the referee suggests. In this study, the LOQ values were set at 10 µg/kg, because it was the lowest concentration that has been validated with acceptable recovery and precision values and identification criteria.

The role of the internal standard is to check the extraction process of the calibration points, which are performed by spiking the blank samples and subjected to the extraction process

Comment: Trueness in assessing the recovery. Please clarify whether blank matrices or real  samples were used for the study. 

Response: As it was described, blank matrices (samples) were used for the study after spiking them with the target analytes at two different concentrations (L249-251, new version) .

Comment: L 310-319 The section describes protocol and should be placed in methods.

Response: The referee is right, and a revision has been carried out throughout the manuscript in order to describe all protocols in the Experimental section.

Comment: What are the LOQ values for each analyte? What are the LOD values? 

Response: As it was stated above, the LOQ values were set at 10 µg/kg, because it was the lowest concentration that has been validated with acceptable recovery and precision values and identification criteria.

We do not know what the reviewer refers to with LOD values. If the acronym means detection limit, in this study they have not been estimated since it is a quantitative method in which its estimation is not mandatory, as indicated for example by the SANTE guide; values ​​lower than LOQ found in real samples are reported as <LOQ. On the other hand, if LOD means limit of determination, as referred to in the European Regulation 396/2005, that means the validated lowest residue concentration which can be quantified and reported by routine monitoring with validated methods. In this respect it can be regarded as the LOQ.

Comment: L 363 Measured samples had levels below the regulatory limits. Explain and justify the accuracy of the measurements, as samples were only detected and were not able to be quantified.  

Response: Effectively, all analyzed samples containing target analytes had concentrations below the MRL (typically higher than the stated LOQ). However, most of the results were found above the LOQ of the validated method and interpolated in the calibration range. So, accuracy of the measurements was appropriate with the use of this previously validated method.

Reviewer 2 Report

The manuscript described the set-up and validation of a new UHPLC-MS/MS method for the simulaneously detection and quantification of 8 phytohormones and 27 acidic herbicides in fruit and vegetables.

The work is interestng and generally well-written.

I suggest a minor English revision.

I highlight that in Table 3 "Analito" is reported instead of "Analyte" 

Author Response

Comment: The manuscript described the set-up and validation of a new UHPLC-MS/MS method for the simulaneously detection and quantification of 8 phytohormones and 27 acidic herbicides in fruit and vegetables.

The work is interestng and generally well-written.

I suggest a minor English revision.

I highlight that in Table 3 "Analito" is reported instead of "Analyte" 

Response: According to reviewer’s indication, English has been revised throughout the manuscript, and analito in Table 3 has been modified by compound in the revised version.

Reviewer 3 Report

The manuscript describes a fast UPLC analytical method for the determination of acidic pesticides and phytohormones in fruits and vegetables. The work carried out is complete, scientifically and technically sound.A massive monitoring programme was also presented, giving an additional value to this method. I recommend its publication after minor revision in the language.

More specifically, there are many linguistic errors that need to be corrected throughout the manuscript. I would recommend to have it read and correct it by a native speaker, since it contains a lot of odd sentences. Some minor edits, just to improve it slightly, as I read it:

  • Line 66: I suggest "It is easy to find in the literature..."
  • Lines 69-70: rephrase "It is rare to find studies in which phytohormones and acidic herbicides are grouped together under one method"
  • Lines 72-73: rephrase the following sentence (syntax error) "Specific characteristics of the target analytes cause that the...."
  • Line 76: I suggest "parent" instead of "original”
  • Line 79: rephrase (syntax error) "It is commonly found extraction methods...."
  • Line 99: Rephrase "During the development of the method was kept in mind the need to reach detection...."
  • Line 105: suggest to delete "full"
  • Line 109: rephrase and simplify "The proposed methodology has been designed giving high relevance to factors such...."

and many more.

Author Response

Comment: The manuscript describes a fast UPLC analytical method for the determination of acidic pesticides and phytohormones in fruits and vegetables. The work carried out is complete, scientifically and technically sound. A massive monitoring programme was also presented, giving an additional value to this method. I recommend its publication after minor revision in the language.

More specifically, there are many linguistic errors that need to be corrected throughout the manuscript. I would recommend to have it read and correct it by a native speaker, since it contains a lot of odd sentences. Some minor edits, just to improve it slightly, as I read it:

  • Line 66: I suggest "It is easy to find in the literature..."
  • Lines 69-70: rephrase "It is rare to find studies in which phytohormones and acidic herbicides are grouped together under one method"
  • Lines 72-73: rephrase the following sentence (syntax error) "Specific characteristics of the target analytes cause that the...."
  • Line 76: I suggest "parent" instead of "original”
  • Line 79: rephrase (syntax error) "It is commonly found extraction methods...."
  • Line 99: Rephrase "During the development of the method was kept in mind the need to reach detection...."
  • Line 105: suggest to delete "full"
  • Line 109: rephrase and simplify "The proposed methodology has been designed giving high relevance to factors such...."

and many more.

Response: English has been intensively revised throughout the manuscript, including all the suggestions indicated by the reviewer.

Round 2

Reviewer 1 Report

 Angel Grande Martínez and colleagues have addressed most of the comments and the authors are now presenting the revised version of the manuscript.  

The abstract and introduction have been improved and the content is much clearer. However, there are still grammatical errors found in the manuscript and a much needed linguistic help is compulsory.

The results and discussion section must be rewritten so that readers and researchers are able to understand the content.

Detailed comments are found below: 

Methods 

Section 2.2 Were the moisture contents in samples of fruit and vegetables determined?  

L 182-183 What was the volume of these solutions that was added to samples? 

The method section has been made very clear for other researchers to follow the protocols. Though, the grammatical errors must be corrected. 

Results and discussion 

L 297 Please include the pH of the mobile phase system after being added with acetic acid. Which acetic acid was added and make sure the acid has also been mentioned in the list of chemicals in the methods section. 

L 311 ‘personalised’ is not an appropriate wording used here and must be replaced. 

L 311-313 the text must be re-written for clarification. As per methods section, it is assumed that samples were ground after purchase and extracted for analysisUnless there was anything that has been mentioned before in the method, authors are advised to be specific about sample preparation. 

L 313 Please be specific with the basic and acid solutions used here by providing molarity and pH, so that other researchers can follow the protocol easily. 

L 319-322 the text is not clearly presented and must be rewritten. 

L 357-360 rewrite the sentence, as such it is not clear at all and difficult to understand what is meant. 

L 376-387 Explain and justify the concentration ranges used for the calibration curves, considering that the MRL for each analyte differs? 

Were lower concentration range prepared to determine lower analytes of interests so that compliance with the regulatory limits can be assessed? 

L 393 Recovery was not associated with a “method that performed well” as claimed in the text. Recovery study was undertaken for in validation of method and this purpose must be inserted in the text. 

Figure 2 which analyte was spiked to the cucumber samples? Please include the compound in the figure caption 

 L 376-387 Explain and justify the concentration ranges used for the calibration curves considering the different MRL values. How were the ranges of standard calibrants were prepared so that the concentrations of the analytes of interests were able to be determined and assessed for compliance with the EU regulation? 

Check the numbering of the tables 

L 459-477 As mentioned in the previous review, the protocol of the samples analyses must be described in the method section rather than in the results/discussion section.  

L 483-485 Rewrite the sentence as such it is quite difficult to understand and delete “two ones” 

Table 4 present the MRL values for tomatoes. 

L 504-506 rewrite the sentence for better understanding 

Author Response

Response to reviewers

Comments: Angel Grande Martínez and colleagues have addressed most of the comments and the authors are now presenting the revised version of the manuscript.

The abstract and introduction have been improved and the content is much clearer. However, there are still grammatical errors found in the manuscript and a much needed linguistic help is compulsory.

The results and discussion section must be rewritten so that readers and researchers are able to understand the content.

Response: All the manuscript has been deeply reviewed in order to improve its English. All the sections has been critically reviewed in order to clarify their contents and facilitate their understanding.

Detailed comments are found below: 

Methods 

Comment: Section 2.2 Were the moisture contents in samples of fruit and vegetables determined?

Response: No, they were not determined. The grouping of samples were carried out according to the classification of commodity according to the SANTE 12682/2019 Annex A.

Comment: L 182-183 What was the volume of these solutions that was added to samples? 

Response: The TTP internal standard was already included into the 10 mL of acetonitrile acidified with formic acid. The text has been modified in order to clarify it to the reader: “Then, 10 mL of acetonitrile acidified with 1 % formic acid and already containing TPP at 0.05 mg/L as a surrogated internal standard were added.” (L. 183 in the new version).

Comment: The method section has been made very clear for other researchers to follow the protocols. Though, the grammatical errors must be corrected. 

Response: All the manuscript has been reviewed in order to correct grammatical errors and clarify its content.

Results and discussion 

Comment: L 297 Please include the pH of the mobile phase system after being added with acetic acid. Which acetic acid was added and make sure the acid has also been mentioned in the list of chemicals in the methods section. 

Response: The apparent  pH of the mobile phase was  ≤5.5). This has been indicated in the new versión of the manuscript (L. 306-307).

The list of chemicals already included the mentioned reagent. However, the list of reagents has been updated including their purity. The sentence has been rewritten as follow: “Formic acid (>99.0% Optima, LC-MS grade), acetic acid (purity higher than 99.8% for HPLC) and sulphuric acid (96% Solution in Water, Extra Pure) were obtained from Fluka.“ (L. 150-152 in the new version of the manuscript).

Comment: L 311 ‘personalised’ is not an appropriate wording used here and must be replaced. Customized custom

Response: The sentence has been rewritten as follow: “(ii) customized addition of water to samples…” following the referee´s suggestions. (L. 324 in the revised version of the manuscript).

Comment: L 311-313 the text must be re-written for clarification. As per methods section, it is assumed that samples were ground after purchase and extracted for analysis. Unless there was anything that has been mentioned before in the method, authors are advised to be specific about sample preparation. 

Response: The text has been rewritten in order to be clarified. In the mentioned sentence, a list of significant differences with methods previously published by other authors is described. The only methodology proposed by the authors (without any modification) is the previously described in sample preparation section. The new sentence says as follow: “In the present study, the QuEChERS extraction method [9] was adapted to extract the target analytes. The main modifications regards to other methodologies previously described were:…” (Lines 321-323 in the current version of the manuscript).

Comment: L 313 Please be specific with the basic and acid solutions used here by providing molarity and pH, so that other researchers can follow the protocol easily. 

Response: It has been updated with the data requested by the referee. The new text is as follow: “…(iii) addition of a lower amount of both, sodium hydroxide 5N (pH ≈ 12) and sulphuric acid 5N (pH ≈ 1) to reduce potential contamination of sample, cost reduction and miniaturization of the method and…” (Lines 326 in the current version of the manuscript).

Comment: L 319-322 the text is not clearly presented and must be rewritten. 

Response: The sentence has been rewritten including the requested data as follow: “In addition, it was observed a reduced matrix effect due to the 2-fold dilution of the obtained extracts. It represented an improvement regard alternative clean-up steps with sorbents (better selectivity) and avoided analyte losses increasing extraction efficiency for most of the target analytes (enhanced robustness).” (Lines 332-335 in the current version of the manuscript).

Comment: L 357-360 rewrite the sentence, as such it is not clear at all and difficult to understand what is meant. 

Response: It has been rephrased as follow: “Analytes were confirmed by mass spectrometry comparing the ion intensity ratios of their most characteristic ions with those obtained for standards analyzed at similar concentrations.” (Lines 367-368 in the current version of the manuscript).

Comment: L 376-387 Explain and justify the concentration ranges used for the calibration curves, considering that the MRL for each analyte differs? 

Response: The sentence has been modified as follow: “This working range was established for all compounds taking into account that: there was good linearity and it included the MRLs in the target matrices” (L 393-394 in the new verion). So, the proposed calibration curves allowed the determination of the compounds at such MRL.

Comment: Were lower concentration range prepared to determine lower analytes of interests so that compliance with the regulatory limits can be assessed? 

Response: The compliance with the regulatory limits could be successfully assessed through the described validation process. Also, as it has been described in the manuscript, the proposed method was satisfactorily accredited under ISO17025 and evaluated using several internal and external quality controls. As previously indicated, the proposed method allows the determination of the target compounds at the regulatory limit for each analyte / matrix combination.

Comment: L 393 Recovery was not associated with a “method that performed well” as claimed in the text. Recovery study was undertaken for in validation of method and this purpose must be inserted in the text. 

Response: We agree with the referee. The sentence has been rephrased as follow: “Recoveries ranged from 86-115% in cucumber samples (Table 2), and from 73 – 120 % in orange commodities (Table 3 - orange) according to the requests of UNE-EN-ISO/IEC 17025:2017 accreditation according to the requirements of the UNE-EN-ISO/IEC 17025:2017 accreditation and SANTE guidelines. “ (L. 410-411 in the current version of the manuscript).

Comment: Figure 2 which analyte was spiked to the cucumber samples? Please include the compound in the figure caption. 

Response: The requested data has been included into the figure caption as follow: “Figure 2. UHPLC-MS/MS chromatogram of: a) blank cucumber sample monitoring transitions at m/z 222/141 (a1) and m/z 222/114 (a2), and b) blank cucumber sample spiked with 10 µg/kg of quimerac and monitoring transitions at m/z 222/141 (b1) and m/z 222/114 (b2).”

Comment: L 376-387 Explain and justify the concentration ranges used for the calibration curves considering the different MRL values. How were the ranges of standard calibrants were prepared so that the concentrations of the analytes of interests were able to be determined and assessed for compliance with the EU regulation? 

Response: It has been previously discussed. All the MRL regulated by the EU for the studied compounds were included into the proposed working range. Therefore, carrying out calibration with such concentration range for all the analytes, it was assessed for compliance with the EU regulations.

Comment: Check the numbering of the tables 

Response: It has been checked and corrected when it was needed.

Comment: L 459-477 As mentioned in the previous review, the protocol of the samples analyses must be described in the method section rather than in the results/discussion section.

Response: The protocol is described in the Method section as the referee recommends. It has been included in the ·2.3 Extraction procedure and sample analysis section, L. 192-199 in the new versión of the manuscript.

Comment: L 483-485 Rewrite the sentence as such it is quite difficult to understand and delete “two ones” 

Response: The sentence has been modified: “In orange, four analytes were detected. Two of those compounds can be used as herbicides or phytohormones, 2,4-D (Figure 4) and dichlorprop (2,4-DP). The other two detected compounds can be used as herbicides, 2-methyl-4-chlorophenoxy acetic acid (MCPA) and triclopyr.” (L. 4504-507 in the new version of the manuscript).

Comment: Table 4 present the MRL values for tomatoes. 

Response: They have been corrected.

Comment: L 504-506 rewrite the sentence for better understanding

Response: The sentence has been rewritten as follow: “The proposed method has been designed to be successfully implemented in testing laboratories to perform routine analyses thanks to its simple sample treatment and rapid chromatographic analysis.“ (L.516-518 in the current version of the manuscript). (L. 527-529 in the current version of the manuscript).
